# Novel Semantic-Based Probabilistic Context Aware Approach for Situations Enrichment and Adaptation

Abderrahim Lakehal [1], Adel Alti [2,*] and Philippe Roose [3]

1    LRSD, Department of Computer Science, Faculty of Sciences, University Ferhat Abbas of SETIF-1, Setif 19000, Algeria; abderrahim.lakehal@univ-serif.dz
2    Department of Management Information Systems, College of Business & Economics, Qassim University, P.O. Box 6633, Buraidah 51452, Saudi Arabia
3    LIUPPA, IUT of Bayonne, University of Pau and Adour Countries, 64000 Anglet, France; philippe.roose@iutbayonne.univ-pau.fr
*    Correspondence: a.alti@qu.edu.sa

**Abstract:** This paper aims at ensuring an efficient recommendation. It proposes a new context-aware semantic-based probabilistic situations injection and adaptation using an ontology approach and Bayesian-classifier. The idea is to predict the relevant situations for recommending the right services. Indeed, situations are correlated with the user's context. It can, therefore, be considered in designing a recommendation approach to enhance the relevancy by reducing the execution time. The proposed solution in which four probability-based-context rule situation items (user's location and time, user's role, their preferences and experiences) are chosen as inputs to predict user's situations. Subsequently, the weighted linear combination is applied to calculate the similarity of rule items. The higher scores between the selected items are used to identify the relevant user's situations. Three context parameters (CPU speed, sensor availability and RAM size) of the current devices are used to ensure adaptive service recommendation. Experimental results show that the proposed approach enhances accuracy rate with a high number of situations rules. A comparison with existing recommendation approaches shows that the proposed approach is more efficient and decreases the execution time.

**Keywords:** ontology; heterogeneous connected objects; situations rules enrichment; situations rules adaptation; situations rules learning

## 1. Introduction

Nowadays, many smart applications such as smart health, smart home and smart city [1,2] require efficient situations enrichment and adaptive service delivery through recommendation systems. The recommendation is the most helpful way to assist users in their everyday activities. The proposed solutions for situations enrichment and service delivery in such applications require low execution time and high accuracy level to meet the user's requirements in a large number of available rules [3].

In the smart environment, ontologies are widely standard models and supporting techniques for improving the recommendation of services and quality of life by detecting the relevant situations of a user and easy access to their related services [4]. However, ontology models can be applied to different smart domains by exploiting their capacities to capture and exchange information through unified view of the data. High accuracy and fastness are the most desired targets for any recommendation ontology-based approach. High accuracy means closing gaps between what users want and available situation's services, while fastness ensures that situation's services are recommended in a short response time. In general, the existing ontologies models use a static mechanism to recommend situations' services. This kind of static recommendation of situations' services results in poor consideration of the whole user's context, thus preventing the dynamic recommendation of relevant situations and long-time adaptation of services to users according to

devices' heterogeneity. In fact, considering rich contextual factors such as the user's role, user's location, day and time, user's implicit and explicit preferences, user's history and current situation during the recommendation process may much better improve the accuracy of the recommended rule items. Furthermore, we aim to assist the user in different moving scenarios by providing pertinent rules/services considering rich contextual factors (e.g., its user's role and location, day and time, user's implicit and explicit preferences, user's history, user's current situation, etc.). For instance, the system will not recommend cinema services or any entertainment services during the weekday although if the user is near to the cinema. However, the system recommends these services on the weekend.

Other approaches have been identified in the literature [4] to recommend the services in the smart environment. Most of them have used content or collaborative approaches, which are based only on user preferences. However, these techniques' performance decreases facing up the complexity of managing a huge number of situations and mobile devices or looking for suitable services while respecting time requirements. For smart environments, we need a roadmap to combine semantic models with uncertain methods to perform the recommendation system according to the accuracy rate and response time. Recently, several surveys and studies in the field of rule-based recommendation [5–27] have focused exclusively on recommendation process techniques regardless of the following questions: what is the context information to take into consideration for a better understanding and an effective result? Which device is the best to use considering the immediate situation? How to develop an efficient recommendation mechanism to inject new situation rules extracted from users' agendas to rapidly filling up the agendas of newly registered users. Most existing recommendation techniques offer relevant rules and suitable services to persons but introduce considerable challenges when it comes to pervasive computing. Existing techniques do not deal with a huge number of situation rules with several moving scenarios (e.g., at home, at work, at car, etc.) and dynamic context evolution exhibiting varying degrees of precision, accuracy and processing time, nor do they handle multiple devices at the run-time. In this scope, to recommend the right situation rule in the right context for the right person, a context-aware semantic model for rule-based recommendation system needs to be considered.

Since situations are correlated with the user's context, it can be considered in designing a recommendation approach to enhance relevancy and reduce the execution time. This paper proposes a new de-facto classification of situation rules that reflect the different types of user's situations rather than dealing with each situation separately, which we believe provides an alternative as a complement to the standard semantic models. More precisely, we define four probability-based-context items (user's location, user's role, user's preferences and time) in their classification. Underlying these selections of such dimensions are three main situations rules classes that should be sufficient to derive other specific situations according to user's role, specific points of days and smart domain. Thus, summary of the contributions of this work are:

- Enhance semantic description of user's context profile and improve classification of situations rule based on four probability-based-context items (user's location and time, user's role, user's preferences and user's experiences).
- Apply multidimensional (*user's context, device capability and rule content*) recommendation space for smart environments.
- Combine semantic classification techniques and Bayesian-classifier to improve recommendation of situation rules and a high accuracy rate by Bayesian-classifier. The weighted linear combination is applied to calculate the similarity of rule items. The higher scores between the selected items are used to identify the relevant user's situations.
- Compare performance between the proposed approach and four other recommendation approaches among the most common algorithms learning-recommendation for various performance metrics.

The remaining paper is organized as follows. Section 2 reviews the existing related works. Section 3 presents and details the proposed recommendation approach. Section 4

presents in detail the proposed methodology. Section 5 discusses some potential results and conclusions. Finally, Section 6 presents the conclusions of our paper and the future research directions.

## 2. Related Works

Recently, recommendation systems have been used in several domains such as movies, tourism e-commerce, health and social network. Recommendation approaches are divided into four major strategies: (i) Content-Based approach, (ii) Collaborative approach, (iii) hybrid approach and (iv) context-aware based approach.

### 2.1. Content-Based Filtering

This type of recommendation approach is based on the analysis of the content of items that have been rated previously by the users and the content of items to be recommended [5]. These approaches aim to recommend similar items to those rated items by a user in the past. They can be used in different domains such as recommending restaurants, hotels, web pages, movies and journals. Many algorithms have been proposed for content-based filtering that predict a numeric value (i.e., *rating of an item*) show how far an item can interest a user.

Tkalčič et al. [6] propose a content-based filtering approach for image recommendation. This approach utilizes affective metadata (i.e., metadata that describes emotions of the user, "this image gives me a sad feeling") and the content of an image to recommend. This work shows that the use of affective metadata in a content-based recommendation system for images yields a significant improvement in the performance of the recommendation much better than using generic metadata (e.g., genre).

Deldjoo et al. [7] propose a content-based video recommendation system that includes a technique to analyze contents of videos and extract videos' features (i.e., low-level stylistic features such as lighting, color, shadows and camera motion). These features can be used to predict intended emotional, aesthetic or informative effects to provide users with recommendations. This technique neglects semantic features such as title of the film, year of production, nationality, genre of video, names of the actors, original language and subtitle. Despite that, the collection of high-level semantic features is more costly because it requires an editorial effort, which is not always available.

To help authors retrieve the relevant journal and speed up the inscription and reviewing processes. Wang et al. [8] propose a content-based recommendation system for journals or conferences publication known as Publication Recommendation System (PRS). This system can suggest the best conferences or journals based on the manuscript summary through a priority order process. The PSR uses two methods including Term Frequency and Inverse Document Frequency (TF-IDF) and chi-square feature selection. The first method TF-IDF offers a way to recognize the important terms of an article by associating a weight for each term [9]. While the second method chi-square statistic calculates the degree of relationships between the term $t$ and a class $c$, such as computer science journals or symposium [10].

Further, the PRS adopts a real-time online system that uses a web tool to automatically update the training model. However, this system achieves 61.37% accuracy for paper recommendations, which is quite low. We believe that accuracy can be improved significantly if the system takes into account other criteria such as authors' preferences, impact factors, the first decision, and special issues.

The major problem in the content-based filtering approach in recommendation systems is the overspecialization problem. In which the system provides recommendations based

only on the user's profile and neglects the user's future interests. The system finds items close to those already recommended by the client. In other words, it does not provide novelty from the user perspective and prevents recommending surprising items that users have not discovered yet.

### 2.2. Collaborative-Filtering (CF)

These approaches are based on aggregating and analyzing users' experiences in the past, then predicting users' interests in the future. As collaborative filtering approaches rely on users' feedback, they can recommend accurately complex items (e.g., video or music content) than content-based filtering. However, the extraction of visual features of videos is quite difficult. In addition, suggesting appropriate items to a client based on their interests and desires extracted from their feedbacks appears naturally easier. One of the major issues in CF is the sparsity of users' ratings.

To address this issue, Zhang et al. [11] present a CF recommendation system based on a slope smoothing scheme and items classification. This approach predicts unrated items by employing two main phases to produce recommendations. The first step calculates the average deviation of two items based on slope one scheme. The second step generates the prediction based on Pearson correlation similarity.

Hernando et al. [12] propose a CF technique based on Bayesian probabilistic model to predict the tastes of users. This technique adapts the factorization of high-rating matrix. Unlike the traditional factorization of matrix, this mechanism assigns a vector k ranging from [0, 1] with an understandable probabilistic model which allows identifying classes of users having similar desires. The new factorization technique offers a significant improvement in terms of quality of prediction and recommendation accuracy. However, this technique imposes a significant computational time during the decomposition of the initial matrix into two matrices.

Recently, deep learning-based methods find a way to submerge in recommendation systems. Wang et al. [13] propose a Neural Graph Collaborative Filtering (NGCF) based recommendation system. This system includes user-item interaction graph into high-order connectivity graph-based embedding function preserving collaborative signal. However, the user-item interaction structure used for understanding user behaviors is quite poor and based only on collaborative signal. More specifically, many other forms of knowledge models can be used to better understand user behavior, such as context-aware model [14–16] and social networks [17]. So far, several collaborative filtering recommendation systems have been studied and tested.

We found that collaborative filtering approaches could address one problem at once either overcoming the cold start (i.e., when a new user has not provided enough ratings, or a new item has not been rated yet, the system is not able to generate reliable recommendations) or augmenting the recommendation accuracy rate. Furthermore, the cold start problem consists of rating data sparsity. The data sparsity evaluates the ratio of available ratings to all possible ratings [18]. In real recommendation systems, the sparsity is very close to one (100%) since users rate a small number of database records [19]. For these causes, it is difficult for collaborative filtering to provide good results addressing these problems. To deal with these problems, researchers use hybrid recommendation techniques where they combine two or more mechanisms, e.g., combine collaborative and content-based techniques [20].

### 2.3. Hybrid Filtering

As we have shown above, each collaborative and content-based technique has its limitations, like the overspecialization, the clod start and the sparsity problem. In fact, hybrid approaches have been proposed to enhance the recommendation accuracy and avoid above-mentioned limitations of collaborative and content-based techniques. Soboroff et al. [21] propose a mixture recommendation technique that depends on both collaborative filtering and content-based filtering to get out the best of both approaches. Content-based

filtering is based on the analysis of the content of past relevant documents to construct user profiles. Furthermore, CF is based on the correlation of users' ratings of past items to recommend novel items where a correlation coefficient is calculated between each user. However, this work neglects the contextual aspect of users such as spatiotemporal-based and role-based user preferences, which reduces the quality of recommendation. Yu et al. [22] develop a platform called a Context-aware Media Recommendation (CoMeR) based on hybrid-processing approach. It combines three different approaches, including, the content-based approach, Bayesian-classifier approach and the rule-based approach. Furthermore, it employs a context-aware model that represents the user profile including, spatiotemporal information, and user preferences. This platform supports media adaptation and recommendations for smart mobile applications. This work shows significant results with accuracy rates around 0.8 and recall rates around 0.75. However, the CoMeR platform lacks flexibility. It provides a static model in which the database is updated in an offline manner. Therefore, we intend to fix this problem by using a learning process classifier that updates automatically and dynamically the database in an online manner. Nilashi et al. [23] develop a hybrid method using multi-criteria-based collaborative filtering for hotels recommendation. Further, to improve the recommendation accuracy of multi-criteria CF, they combined three techniques, including, Expectation-Maximization (EM) algorithm, Adaptive Neuro-Fuzzy Inference System (ANFIS) and Principal Component Analysis (PCA) for dimensionality reduction. This work shows high accuracy on the TripAdvisor dataset. However, items' model and users' model are updated in an offline manner, which makes this work incapable to incrementally adapt new data ratings. Therefore, incremental learning is needed to consider new updates on the fly. Services recommendation becomes an important aspect in pervasive environments due to their dynamicity and heterogeneity. Various recommendation approaches that we have studied in previous sections focus only on rating and similarity measurement and neglect context information. Neglecting the context in recommendation systems remains inappropriate for such dynamic environments (i.e., context changing). Therefore, it will be necessary to consider context-aware approaches in recommendation systems to recommend more relevant services for the user's needs and usage context.

### 2.4. Context-Aware Recommendation System

Recently context has been widely used in multiple disciplines [10]. For instance, classic recommendation systems can take advantage of contextual information such as location, time, user activity and social information, to improve recommendation accuracy. The integration of context information has been adopted by many researchers in many areas including pervasive environments, marketing, smart domains and IoT [10].

Some researchers exploit the time and location information to reduce the search space that enables to enhance recommendation accuracy, namely, time-aware and location-aware recommendation [24,25]. Point-Of-Interest (POI) recommendation is one of the most context-aware recommendation systems that is based on spatiotemporal context information. It recommends the best places to visit. Yuan et al. [15] introduce a spatiotemporal-aware POI recommendation system to recommend a list of POIs for a given user's location at a specific time, e.g., recommend for a user to visit a nearby restaurant at midday. They develop a POI recommendation model based on human behaviors, namely, temporal behaviors and spatial behavior. Moreover, temporal and spatial behavior can play a significant role in analyzing users' activities and improving recommendation accuracy. For instance, if many users visit a restaurant at midday but very few users visit a library at the same time, then the restaurant should be given a higher priority than the library during recommending POIs for a user to visit at midday. However, this work tends to produce a pool of unrelated POIs that the user may not continuously visit due to the lack of related associations between POIs. To cope with that issue, Zhou et al. [16] propose a recommendation system based on user-effective Point-of-Interest path (POI) that can recommend POIs by considering both possible associations and diversity features of POIs. In that case, they develop a top-k POI

recommendation model based on effective path coverage to improve the performance of the recommendation algorithm.

People in real life tend to seek advice from their entourage based on social context before trying or purchasing something new. Integrating social context in recommendation systems increase significantly the performance of the recommendation in terms of accuracy. Researchers have used social information (e.g., family, followed and followers, friends list, relationships, trusted and untrusted users) to improve the prediction when providing recommendations. Hong et al. [17] propose context-aware recommendations using a role-based trust network. They use the term *role* to model common context-aware interests within a group of users. The user can play different roles (e.g., diner, movie fan and reader), which may change dynamically by context changes. They use Weighted Set Similarity Query (WSSQ) algorithm to calculate the value of trust between two users in order to build the user's role-based trust network in a given context. The recommendation is made based on both calculated user's role-based network and user-item rating matrix. However, user context and role sets are predefined in an offline manner, which makes this work incapable to incrementally adapt to new context users. Young et al. [18] propose a personalized recommendation system based on friendship strength using Twitter as a source of big Social Networks Systems (SNS). It recommends interests to users by considering various characteristics of big SNS. This approach is based on social information for measuring closeness between users that are defined as friendship strength. Friendship strength is calculated using three categories of social data, including contents generated by users, relationships information and interaction information. More precisely, three similarities are calculated for each social data category, including personal, group and interaction similarity, where friendship is the combination of these three similarities. However, this work ignores user context and may suffer from the cold start problem especially when the user has few friendships in his social circle. We also cite a method based on a collaborative social environment and external providers. Wen et al. [26] propose a recommendation system for mobile applications based on rule items and context information. A semantic model is employed to define correlations among context information. They use a probability model to predict user's interests based on the semantic model. Currently, Karchoud et al. [27] propose a proactive injection mechanism to inject new situations from external context sources into long-life application. This mechanism uses a collaborative social environment to generate a user-friendly situation model to inject new situations considering user's context information.

### 2.5. Comparison of Recommendation Works

In recent years, various recommendation techniques [5–27] have been developed to improve accuracy, adaptation and flexibility using different filtering techniques and strategies. Table 1 illustrates a comparison between several approaches in terms of accuracy, online learning process, data set, scalability, adaptation and flexibility. We have concluded that the content-based recommendation systems [5–10] support the cold start problem in case of updating the data set with new items. However, they suffer from low accuracy compared to collaborative filtering systems. Collaborative filtering techniques [11–20] provide a better accurate recommendation but suffer from the cold start problem. The existing hybrid approaches [20–22] aim to combine different recommendation approaches to yield better recommendations across the board. However, these approaches are not flexible against context changes and evolution of users' needs. In other words, they neglect context-awareness during the recommendation process. Considering context information during the recommendation process improves the recommendation accuracy. Thus, several recommendation approaches adopt context awareness [23–27] to achieve a high recommendation accuracy. To cope with this limitation, the purpose of this research paper is to present a novel approach. It consists in enriching the user's agenda with relevant situations rules. We have the opportunity to classify situations rules by role, location and time in

order to guide and filter information depending on their context and preferences to adapt the search process to the specific needs of users.

**Table 1.** Comparison of recommendation approaches.

| Filtering Approaches | | Accuracy | Online Learning | Dataset | Scalability | Adaptation | Flexibility |
|---|---|---|---|---|---|---|---|
| **Content-based** | [6] | 63.00% | ✗ | images | ✗ | ✗ | ✗ |
| | [7] | 70.50% | ✗ | Movies | ✗ | ✗ | ✗ |
| | [8] | 72.30% | ✗ | Sc. papers | ✗ | ✗ | ✗ |
| **Collaborative-based** | [11] | 80.70% | ✗ | Movies | ✓ | Partial | ✗ |
| | [12] | 84.60% | ✗ | Netflix | ✓ | Partial | ✗ |
| | [13] | 84.60% 97.30% | ✗ | Gowalla Book | ✓ | Partial | ✗ |
| **Hybrid-based** | [21] | 83.50% | ✗ | Movies | ✓ | Partial | ✗ |
| | [22] | 80.00% | ✗ | Movies | ✓ | ✓ | Partial |
| Context-aware — Trust-aware | [18] | 87.84% | ✗ | Services | ✓ | ✓ | Partial |
| Context-aware — Location-aware | [24] | 64.50% | ✗ | Services | ✓ | ✗ | Partial |
| Context-aware — Time-aware | [25] | 43.40% | ✗ | Services | ✓ | ✗ | Partial |
| Context-aware — Spatiotemporal-aware | [15] | 51.00% | ✗ | Foursquare | ✓ | ✗ | Partial |
| Context-aware — Role-based | [17] | 63.50% | ✗ | Epinions | ✓ | ✗ | Partial |
| Context-aware — Environment-aware | [28] | - | ✗ | Tourism | ✓ | ✓ | ✓ |
| Context-aware — Mobile-aware | [26] | - | ✗ | Tourism | ✓ | ✓ | Partial |

## 3. Situation-Based Contextual Model: Definitions and Formalizations

Situations are increasing in unprecedented ways in different areas of pervasive environments. Users can be found in different situations surrounded by various mobile devices, such as personal mobile phones to accomplish their daily tasks. In practice, the specification of a huge number of situations for each user is quite difficult. However, previous studies [27,28] reveal that the recommendation of the right situation rule in the right context is also a real problem for users, even beyond the users' experiences.

Therefore, it is necessary to recommend useful rules through a recommendation process according to the available devices with current context to automatically enrich user's agenda. In this context, three approaches can be used, which are content-based filtering, collaborative filtering and hybrid filtering [28]. In this work of recommendation of relevant situation rules, ontology and hybrid filtering with Bayesian-classifier can play a crucial role to implement context-aware rules classification that enhances relevancy of suggested rules. Realizing ontology-based probabilistic system for context-aware situations enrichment and adaptation in smart environments gives two main benefits for both semantic and relevance recommendation. The first benefit is achieved through the generic and extendable ontology model Multi-OCSM (Multi-layered Ontology-based Composite Situation Model) [28] to unify the representation of situation rules as well as the sharing and classification of context rules. Multi-OCSM is a formal context and situation-based ontology that can play a vital role in facilitating reasoning by formally representing and reusing smart domain knowledge. We used Multi-OCSM to classify user's situation rules (e.g., role-based, localization-based and rule-based on localization and role) before the recommendation process to achieve a high quality of recommendation considering three context categories (user preference, situation context and device capability). We aim to achieve a modular and extensible ontology, additional fields should be easily incorporated according to the characteristics of the examined smart domain. The second benefit is attained through the combination of user explicit and implicit preference, situation context, and device capability with the ultimate goal of covering a multidimensional recommendation space and performing the recommendation system. We detail in the next section the multidimensional recommendation space of the proposed recommendation approach.

### 3.1. Modeling Multidimensional Recommendation Space

In this section, we present a new hybrid recommendation model based on three dimensions categories as input: (1) user context dimension including location, time, role and preferences, (2) device capability dimension including CPU speed, memory size and network bandwidth and (3) rule item dimension as rule's content. These input dimensions are the base to develop the recommendation system intended to suggest useful and relevant situations rules to the user based on his context (i.e., user location and user roles) and his device capabilities. The dimension of user context reflects different spatiotemporal information related to prior usage rules as well as the assigned rules scores, information related to possible roles that a user can play in the smart space (driver, student, etc.), and information related to user's explicit and implicit preferences. The rule item dimension consists of several projections on different context attributes. Each projection is a combination of location, time, role, etc. The device dimension defines device's features and capabilities. The score of rule item is defined as an output. Since the score determines the user's interest level in the suggested situation rule item, it is given from a lower value (0) of the score (least preferred) to a higher value (1) of the score (most preferred) (see Figure 1). The score is calculated as presented in the following relation:

$$R(UserContext, \ RuleItem, \ DeviceCapability) = score$$

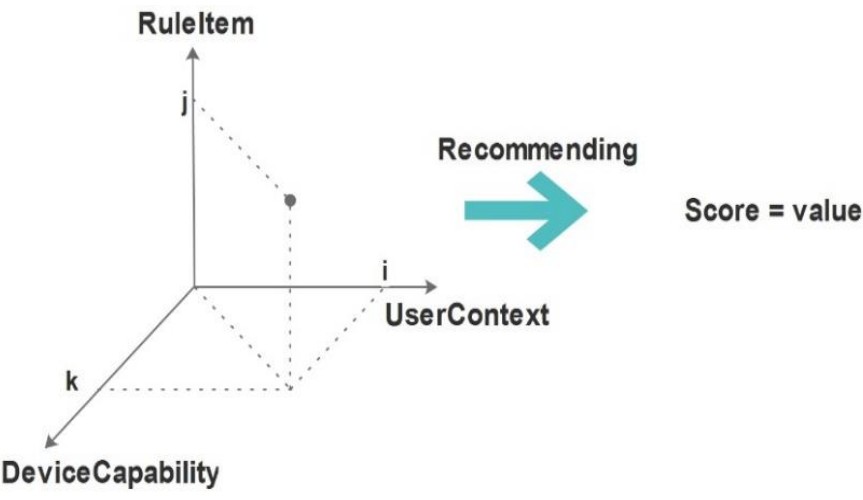

**Figure 1.** Proposed multidimensional recommendation space.

We consider as inputs:

- $UserPreference\ (i) = \{(security; \ 0.65), \ (health; \ 0.41), \ (home; \ 0.58), \ (work; \ 0.41)\}$.
- $RuleItem\ (j) = \{(domain; "home"), \ (day; "weekday"), \ (field; "security")\}$.
- $DeviceCapability(k) = \{(CPU; 4.5\ \text{GHz}), \ (RAM; 4\ \text{GB}), \ (Modality; [text, \ audio, \ image, \ video])\}$

We obtain as output: **Score = 0.88.**

### 3.2. The User Context Profile Formalization

The user profile includes information about the user, such as location, time, agenda and role. User's roles may be "*student*", "*citizen*", "*driver*", etc., and may be determined dynamically using user's spatiotemporal information and agenda:

$$\exists\ l, \ t/agenda(location(l); time(t)) \rightarrow Role_i$$

In a particular case, only the agenda and time can be used:

$$\exists \, t/agenda(time(t)) \rightarrow Role_i$$

### 3.3. The Rule Preference Formalization

The rule preference indicates the user's explicit and implicit preferences regarding the rule content. It is defined by the pair $\langle term, \; weight \rangle$. The *weight* is between $-1$ and $1$ reflects the level of the user's interests in such terms. For instance: $\langle Home; \; 0.75 \rangle$ ; $\langle Security; \; 1 \rangle$ ; $\langle Health; -0.75 \rangle$.

### 3.4. The Device Context

The available device resources represent the device context (i.e., memory size, network bandwidth and protocols).

### 3.5. The Situation Rules Formalization

In this formalization, we should model any contextual application by a set of situations. They are built by a set of events that occurred at a particular location and time as well as set of rules. These rules are used to check these events, and thus inferring the situation. The user defines a set of daily situations that are detected by situation rules. A situation rule is defined by several projections; it has different axes (e.g., location, time, role, sensor value, etc.). We flatten the axis into primitives (e.g., *inside*, *outside*, *near* and *far* for location) to enhance the rule's description. Furthermore, rule modeling can include various exceptions as new dimensions. In addition, we need to determine what the user checks when this situation occurs. Therefore, by considering the relationships between a situation and actions, the actions are activated based on identified situation. The textual description of situation rule is defined as follows:

$$R : Name \, ; \; P_1[Axis(Primitive(value, \; tolerance)\ldots);\ldots]$$

$$P_n \, [\ldots] \; EXCEPT[Situation/Axis(\ldots) \, ] \rightarrow$$

$$Actions[Action_1, \; Action_2, \; \ldots]$$

### 3.6. Classification of Situation Rules

The content of situation rules have different nature and can be defined for any smart domain, which we classify them as follows:

- **Role-based rules:** This kind of rules depends on the user's role. As an example of role-based rules, Rule#1(*Work)* rule checks time axis is equal to the *planed time* and user's job is an *employer*. This rule is described as follows:

$$Rule\#1 \; : Work \, ; \; P_1[Time(PlnedTime(\text{work}));Role(is(employer))]$$

$$EXCEPT[Situation(vacation) \, ] \rightarrow$$

$$Actions[deploy \; work \; space, \; put \; the \; phone \; silent]$$

- **Localization-based rules:** This kind of rules depends on the user's location. As an example of localization-based rule, Rule#2 (*Alarm home*) situation rule consists of two projections: *P*1 and *P*2. *P*1 checks the user's location is outside home and *P*2 checks time is between 8 pm and 5 am. It is, exceptionally, unchecked in Sunday. It is described as follows:

$$Rule\#2 \; : Alarm \; home \, ; \; P_1[Location(Outside(Home, \; 20))] \; OR$$

$$P_2[Time(After(8pm)Before(5am);Location(Inside(Home, 10)]$$

$$EXCEPT[Time(Sunday) \, ] \rightarrow$$

$$Actions[deploy \; security \; alarm \; application]$$

- **Rules-based on localization and role:** These kinds of rules depend on both location and role of a user. For example, Rule#3 (*Meeting*) situation rule checks the user's location is inside the meeting room and time axis is equal to the meeting time and his role *axis* is employer.

$$Rule\#3 : Meeting ; P_1[Location(Inside("MeetingRoom", 2))$$

$$Time(PlnedTime(Meeting)); Role(is(employer))] \rightarrow$$

$$Actions[deploy\ meeting\ application]$$

## 4. Proposed Approach

The context is crucial and has a serious impact on the entire situation recommendation process. We propose a novel semantic-based probabilistic system that enriches and adapts dynamically user's situation rules in vast domains (city, work, shopping, travel, tourism, etc.). The objective of the situation enrichment system is to provide suitably adapted rules to users. These rules have to fit with users' preferences, users' current situations, and device's capabilities. Figure 2 exhibits the general architecture of the proposed recommendation system. This system is described in two main processes including respectively the learning process and the recommendation process.

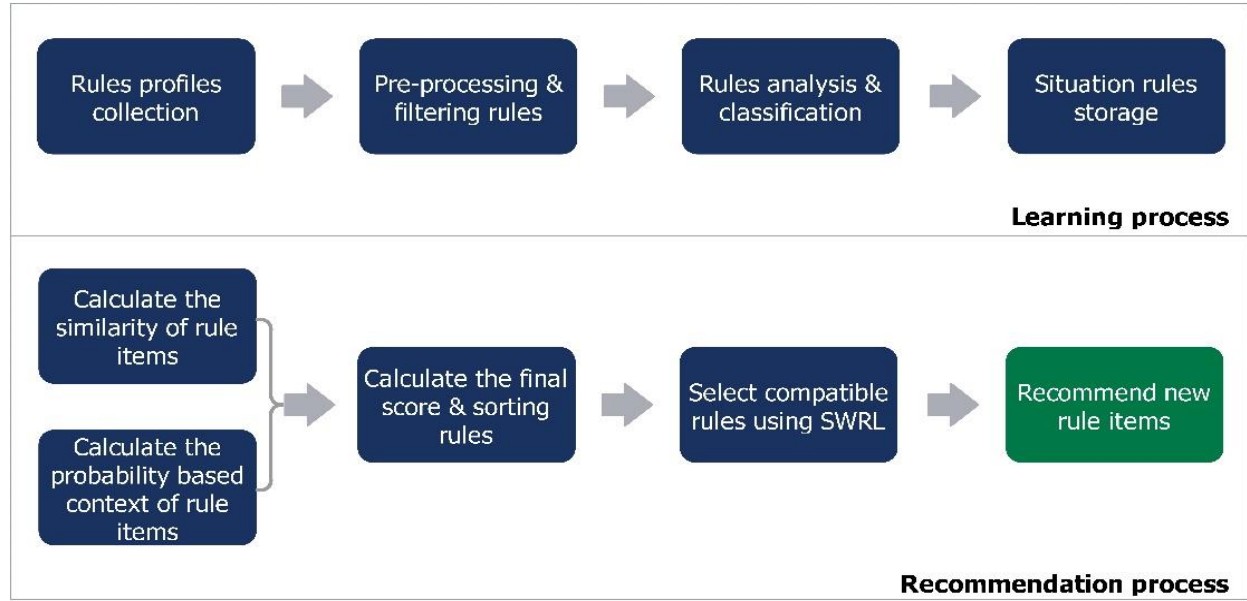

**Figure 2.** General architecture of the framework.

The learning process classifies situations rules according to their content after applying pre-processing and filtering techniques. It is based on four main phases, (1) rules profiles collection, (2) pre-processing and filtering rules, (3) rules analysis and classification and (4) situation rule storage in rules metadata repository. The recommendation process calculates the final score and recommends new rule items for the user. It is based on five main phases, (1) calculate the similarity of rules items, (2) calculate the probability-based context of rules items, (3) calculate the final score and sorting rules, (4) select compatible rules using SWRL (Semantic Web Rule Language) and (5) recommend new rule items. In the next section, we will highlight each part's different steps of the proposed recommendation system.

*4.1. Situation Rules Learning Process*

The proposed learning process classifies situation rules according to semantic rule ontology. The learning process takes a collection of situations rules and outputs classes of learned rules. It consists of the following four main steps (Figure 3).

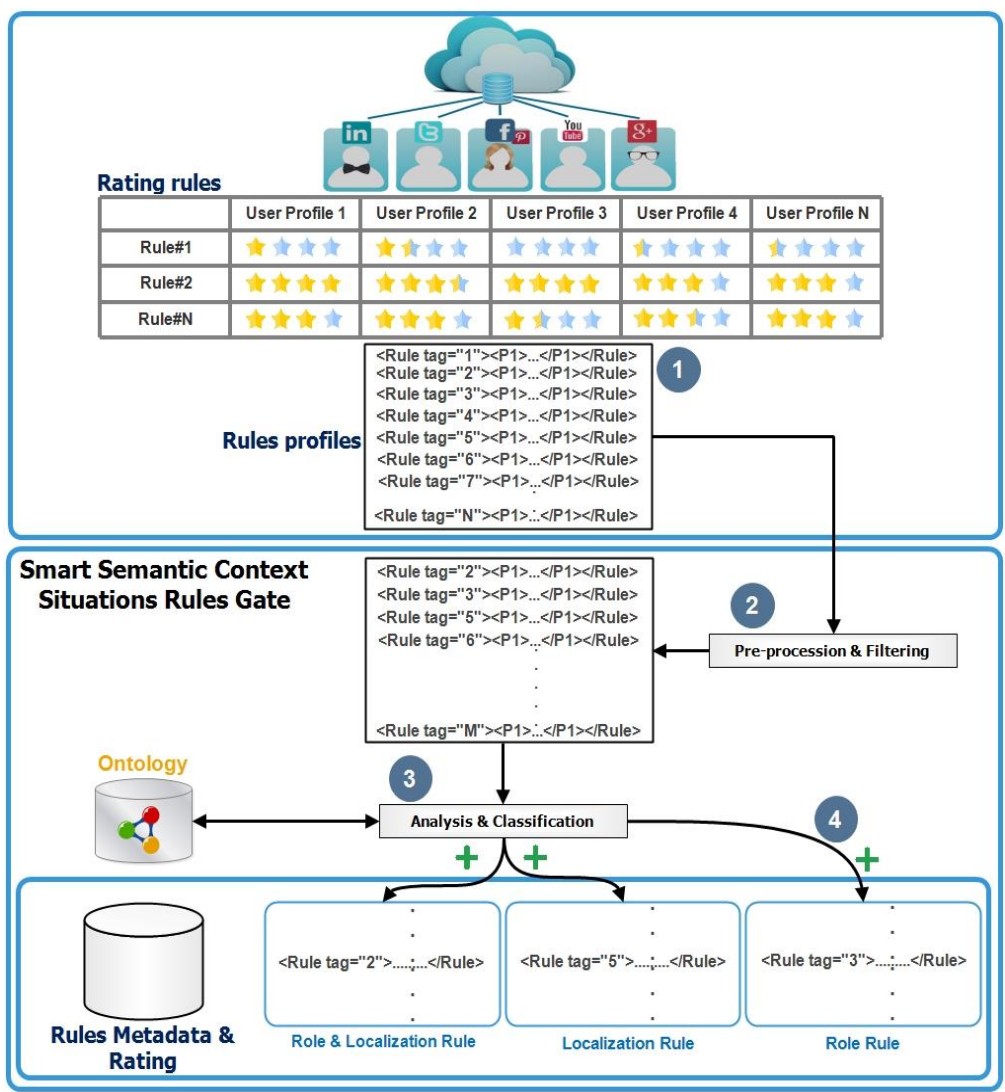

**Figure 3.** An overview of situation rules learning process.

**Step 1—Collection of rules profiles:** consists of collecting situation's rules from vast platforms and social media then sorting them according to users rating from the most popular until the less popular. After the collection of new situations, the system sends them to the preprocessing module for filtering and prepossessing.

**Step 2—Processing and filtering rules profiles:** this consists of eliminating redundant and insignificant rules. For instance, rule without consequence and rule without antecedent.

**Step 3—Ontology-based rules analysis and classification:** We analyze the situation rules with the defined classes in MULTI-OSCM ontology to decide in which class we classify them. From situation rules, we extract a list of semantic context data relates to user's role, user's location, or both from WordNet [14]. For user's location, WordNet gave semantic data, such as Home, Work, University, Office or Car. For user's role, WordNet gives semantic data, such as Citizen, Worker, Student or Driver. We define three classes that represent generic context rules according to the location and the role of user: role and location-based, location-based and role-based. A given situation rule is classified into one of these generic classes based on the location of a user (Home, University, Office,

Hospital, Car, City, Parking or Market) and his role (Citizen, Student, Worker or Driver). If the condition of rule contains both location and role terms, then we classify it into role and location-based classes. Otherwise, we check if it contains only location terms, so we classify it into location-based classes. If not, we classify it into a role-based class. Then, we determine the most popular situation rules based on the users' ratings. The user's rating is ranging from 1 (useless rule) to 5 (excellent rule). Each user can give his opinion about a given situation rule. The most popular situations for the user's current locations and role could be recommended. The popularity of a situation rule is calculated using the following formula:

$$PS_i = \frac{\sum_{j=1}^{N} Rating\_S_i^j}{N} \tag{1}$$

where:

- $PS_i$: the popularity of situation *i*.
- $Rating\_S_i^{U_j}$: the rating of user *j* for situation *i*.
- *N*: the number of users rated the situation *i*.

  **Step 4—Storage of situation rules**. Each situation rule is stored in their appropriate class.

### 4.2. Recommendation Process of Situation Rules

This section presents a hybrid recommendation strategy for situation enrichment and adaptation based on a weighted linear combination equation. This latter, is a combination of two approaches, the first consists of rule's content-based semantic similarity, while the second is a Bayesian-classifier. The objective of the recommendation process is to provide suitable rules to users. It takes the user's preferences, situational context and device information as input to calculate the situation rules' scores and returns the relevant rules ranked according to their corresponding final scores. On the one hand, a content-based recommender deals only user's preferences and rule item content. On the other hand, we use Bayesian classifier to evaluate rule items regarding user situation context. The proposed recommendation process operates in the following six main steps (Figure 4):

**Step 1—Calculation of rule semantic similarity based on content.** After the learning process, in the first phase of recommendation, the sorted rules will be used to calculate the similarity between user preferences and rule items. It compares the content of the rule item and the terms of user's explicit and implicit preferences.

**Step 2—Calculation of rule probability based on situational context.** In the second step of recommendation, we compute the probability for recommending the rule item knowing that we take into account the user situation context.

**Step 3—Calculation of rule final score and sorting rules.** In the third step of recommendation, we calculate the final score of each rule item including explicit/implicit preferences and Bayesian classifier. Then, we use the final score in order to sort the rule items from the highest score to the lowest.

**Step 4—Selection of compatible rules based on the device's capability.** For running and adaptation phases, the system selects compatible rules based on the device's capability through an inference engine. It infers the situation rules, which have the minimum requirements (e.g., memory size, CPU and storage) of available devices (e.g., smart TV, smartphone, tablet, actuator, etc.) in user's domain (e.g., home, office, car, etc.) and, finally, add them to the user agenda.

**Step 5—Recommendation of new rules.** In this step, we take recommended rules with the highest scores and update both the ontology model and the user's agenda based on the user's needs.

**Step 6—Agenda and ontology update.** In the sixth step, we create a semantic link between the user's agenda and situation rules stored in the ontology model. When the user's situation is evolved, the recommended list is changed accordingly.

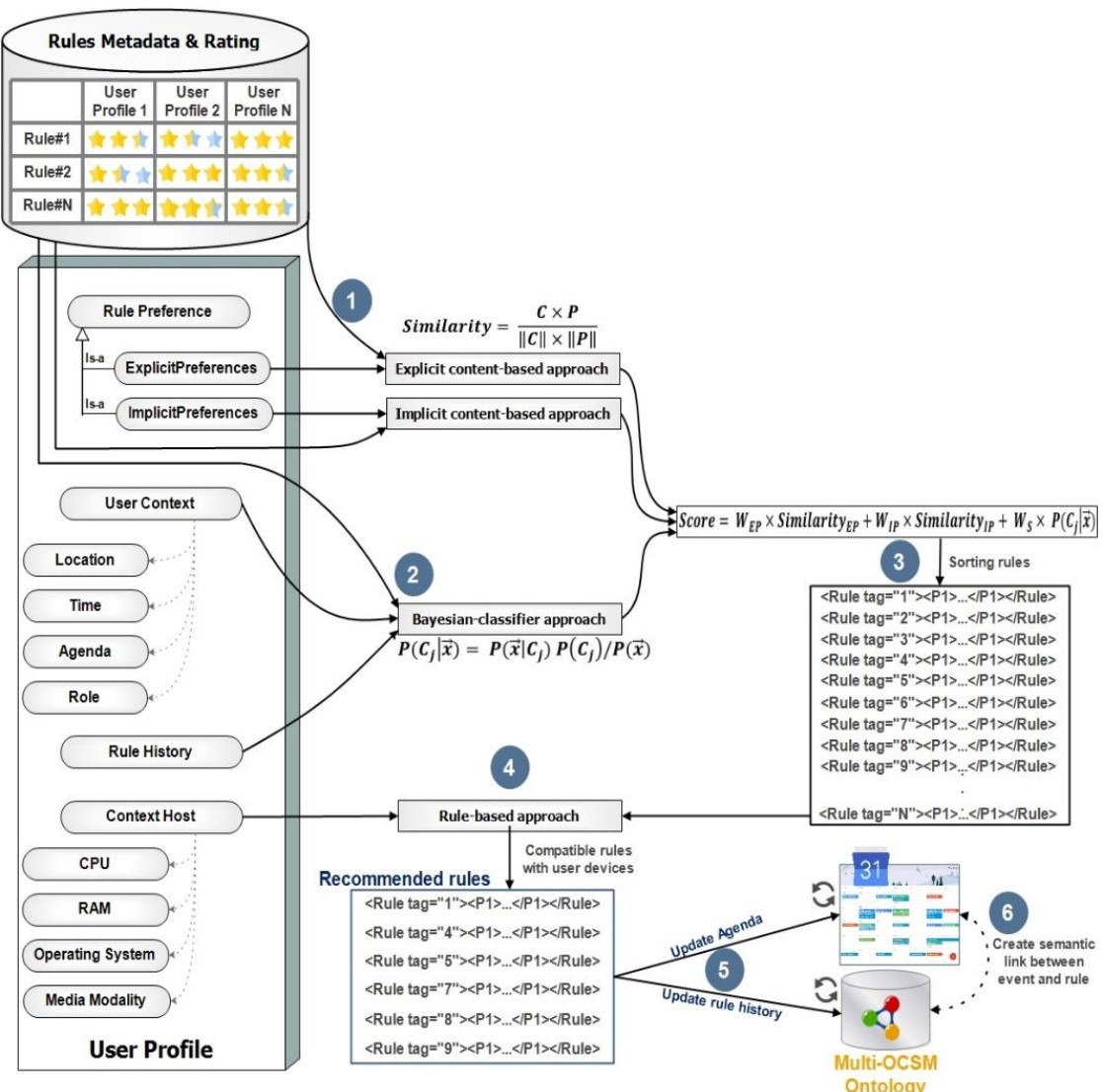

**Figure 4.** Hybrid rule-based recommendation process.

### 4.2.1. Rule's Content-Based Approach

A situation rule is a hierarchical structure of specific depth, which consists of terms, mostly called tags. In smart domains, tags are associated with different fields (security, demotics, work, shopping, tourism, etc.) for efficient rules discoverability. The methodology of calculating the degree of interest of a rule item regarding the user context is presented in this section. However, before the quantification of this degree of interest, some steps should be followed beforehand. Specifically, each user defines explicit preferences in which they will be specified as a vector $EP = (CP_1 \dots CP_n)$, where $CPi$ is configuration preference. Each $CP$ contains four features:

- **Domain:** The application's domain in the configuration preferences, as a string of the possible values (Home, Car, Office, University, Shop, Security, Hospital).
- **Day:** The days in the configuration preferences, as a string of the possible values (Weekday, Weekend).
- **Time:** The time in the configuration preferences, as a string of the possible values (Morning, Afternoon, Evening, Night).
- **Field:** The field of user's domain in the configuration preferences, as a string of the values (Security, Demotics, Study, Health, Work, Shopping).

The preferences of the users can be specified explicitly using GUI or extracted implicitly from his rules. The explicit preferences are described using XML and stored in file called *configuration preferences*. Users can define many configurable preferences (i.e., domain, day, time and field). Besides, the user can specify the range of rules for the recommendation in terms of rules' popularity (i.e., *low, medium and high popularity*). The vector of terms $EP\_Term = (t_1 \ldots t_m)$ is extracted from the vector $EP$, as defined in Algorithm 1.

---

**Algorithm 1: Extract Terms from Explicit Preferences (EP).**

**Input:** Explicit preference vector $EP = (CP_1 \ldots CP_n)$;
**Output:** Term vector $EP\_Term = \varnothing$;
**Begin**

1     : **For each** element $CP_i$ from $EP$
2     :    **For each** term $t_j$ from $CP_i$
3     :        **If** $(t_j \notin EP\_Term)$ **Then**
4     :            $EP\_Term \leftarrow EP\_Term \cup t_j$;
5     :        **End If**
6     :    **End For**
7     : **End For**
8     : **Return** $EP\_Term$

**End**.

---

For each pairs of user-item, we compute weights $w_i$ regard an explicit term $t_i$ in order to get the normalized value as follows:

$$w_i = \frac{number\ of\ occurrence\ of\ term\ t_i\ in\ EP}{e^{|EP|}} \tag{2}$$

Similarly, the vector of terms $IP\_Term = (t_1 \ldots t_m)$ is extracted from the Rules Repository (RR), as defined in Algorithm 2. For each user as a vector $RR = (r_1, \ldots, r_n)$, where $r_i$ is a rule's metadata. The weights $w = (w_1, \ldots, w_m)$, where $w_i$ is the weight of implicit term $t_i$ are calculated as presented in Equation (3).

---

**Algorithm 2: Extract Terms from Rules Repository (RR).**

Input: Rules repository vector $RR = (r_1 \ldots r_n)$;
**Output:** Implicit preference vector $IP\_Term = \varnothing$;
**Begin**

1     : **For each** element $r_i$ from $RR$
2     :    **For each** term $t_j$ from $r_i$
3     :        **If** $(t_j \notin IP\_Term)$ **Then**
4     :            $IP\_Term \leftarrow IP\_Term \cup t_j$;
5     :        **End If**
6     :    **End For**
7     : **End For**
8     : **Return** $IP\_Term$

**End**.

---

$$w_i = \frac{number\ of\ occurrence\ of\ term\ t_i\ in\ RR}{e^{|RR|}} \tag{3}$$

The aforementioned parameters, namely term and weight, are of generic nature and can be calculated for any smart domain. The terms are not equally important (i.e., *by their*

*nature, terms in the location or field tag might be more important than subfield tag*), so we assign important factors to give each term's relative importance for rule item metadata.

The server stores a description of rules history, which are described using XML. Tags are associated with location, time, role and fields for better rule content organization and discoverability. Obviously, some tags are more important than others. The vector weight $W = \{W_x \mid x \in S\}$ is used to define that a tag $x$ has their relative importance (assuming $W_{location} = 0.75$, $W_{role} = 0.45$, $W_{field} = 0.75$, $W_{subfield} = 0.35$).

We define two vectors called Preference Pair: $PP = ((term_1, weight_1) \dots (term_m, weight_m))$ based on the weight $w_i$ of term $t_i$ of each user preference (implicit and explicit). We also define the user preference $P = (w_1 \dots w_m)$, based on the weight values of each $PP$ vector. Similarly, each rule item is presented as a vector $C = (u_1, u_2 \dots, u_m)$ containing weights $u_i$ of term $t_i$. The vector $C$ is generating using Algorithm 3.

The similarity measure between the rule item and user preference is calculated as presented in Equation (4). It should be noted that the similarity between the rule item and user preference in this work is the cosine similarity since it projects the rule items of the same context dimension in the 3-D recommendation space closer together.

---

**Algorithm 3: Generating the Vector C.**

**Inputs:** Rule item's metadata $R$ (item to recommend);
The vector $PP$;
**Output:** The vector $C$;
**Begin**

```
1     : For each term value t_i from PP
2     :       If (t_i   is merely included in tag x of R) Then
3     :       u_i ← W_x;
4     :       If (t_i   is included in two or more tags of R) Then
5     :       u_i ← Max{W_x};
6     :       If (t_i   isn't included in any tag of R) Then
7     :       u_i ← 0;
8     : End For
9     : Return C
```

**End**

---

For instance, location item of user context dimension: university, faculty and study room are oriented closer together in 3-D recommendation space. The closer meaning of context terms, the higher is the cosine similarity. The similarity measure is calculated as:

For each pair of user-items, we compute weights $w_i$ regard an explicit term $t_i$ in order to get the normalized value as follows:

$$Similarity = \frac{C \times P}{\|C\| \times \|P\|} = \frac{\sum_{i=1}^{m} u_i w_i}{\sqrt{\sum_{i=1}^{m} u_i^2 \sum_{i=1}^{m} w_i^2}} \tag{4}$$

### 4.2.2. Bayesian-Classifier Approach

The main idea of the proposed approach is to use Bayesian classifier to evaluate rules according to current situation context after classifying them in groups using situation multidimensional model. The Bayesian approach is one of the most frequently used classification methods. It considers flexibly uncertain context through ontology model and enables us to adapt to the changing context. For instance, the *situation* (*home*, *weekday*, *morning*, *citizen*) can be represented as a class. The situation context classification can be done on four dimensions. These dimensions are defined as follows:

— **Location:** The user's location has the possible values (Home, University, Office and Outdoors).
— **Day:** The days have the possible values (Weekday and Weekend).

- **Time:** The time has the possible values (Morning, Afternoon, Evening and Night.) and,
- **Role:** The user's role has the possible values (Citizen, Student, Worker and Driver).

As it can be seen, *day* and *time* dimensions are applicable to all smart domains, whereas *role* is suited to specific smart domains (e.g., *Home, University, Office and Car*). Since Cartesian product for the 4-dimensions, *location* × *day* × *time* × *role*, equal to $4 \times 2 \times 4 \times 4 = 128$ *classes* has been covered, all possible probability of classes can be calculated. This probability should not depend merely on the user's location (*home, university, office and outdoors*) of a tag, even if that location is large enough and the tag is associated with numerous locations. Additional dimensions which should also be considered are *day* and *time*. The role dimension is introduced to determine the role of user. In this case, values of the *role* denote greater importance in determining probability of a rule item for a certain class. Other key tags that need to be considered are the value of the "Filed and sub-Field" of each rule item, as it cannot only be used to identify relevant or irrelevant item rules but also to balance the aforementioned role values. For instance, if we considered the class $C_i = (home, weekend, evening, citizen)$, then we can compute the probability of a rule item $\overrightarrow{x}$ of a given class $C_i$, as defined in Equation (5).

$$P(C_j|\overrightarrow{x}) = P(\overrightarrow{x}|C_j)\, P(C_j)\, / P\left(\overrightarrow{x}\right)$$
$$P\left(\overrightarrow{x}\,|C_j\right) = \prod_{i=1}^{4} P(f_i|C_j)$$
$$P(C_j) = k(C_j)\, / T \tag{5}$$
$$P\left(\overrightarrow{x}\right) = \sum_{j=1}^{k} P\left(\overrightarrow{x}|C_j\right) P(C_j)$$
$$P(f_i|C_j) = \frac{n(f_i,\, C_j)\, + 0.5}{n(C_j)\, + 0.5|V_j|}$$

where

- $f_i$ represents the *i*th tag *of* $\overrightarrow{x}$.
- $k(C_j)$ represents the total number of users that has used the situation rules in $C_j$.
- $T$ represents the total number of rules in situation rules repository.
- $n(f_i,\, C_j)$ represents number of tag $f_i$ appears in $C_j$.
- $n(C_j)$ represents the sum of all tags appear in $C_j$.
- $|V_i|$ represents the total number of tags appearing in $C_j$.

For example, we can calculate the probability of adding a rule item (house security's field) in an outdoors repository on a weekday. At last, the final score of the rule item will be measured using both content-based approach and Bayesian-classifier approach. The final score of the proposed recommendation process is calculated by the weighted linear combination as defined in Equation (6).

$$Score = W_{EP} \times Sim_{EP} + W_{IP} \times Sim_{IP} + W_S \times P(C_j|\overrightarrow{x}) \tag{6}$$

where

- $W_{EP}$ is assigned to explicit similarity $0 \leq W_{EP} \leq 1$.
- $W_{IP}$ is assigned to implicit similarity $0 \leq W_{IP} \leq 1$.
- $W_S$ is assigned to situation probability similarity $0 \leq W_S \leq 1$.
- $W_{EP} + W_{IP} + W_s = 1$.

These weights reflect the relative importance of preferences and contextual situations. The obtained scores using the equation given above will be used to rank rule items. These items will be checked and deployed on available devices, sensors and actuators of current user's domain.

### 4.2.3. Rule-Based Adaptation Process

The proposed recommendation approach is applied on limited mobile devices of a user domain in order for the "selected recommendation rules" to be checked. In addition, we resolve the problem between resources characteristics (devices that run-rule items) and required characteristics (rule item checked and adapted). Examples of such attributes include CPU charge, sensor availability and memory size. Because of this, the capability context is necessary for maintaining the compatible services to improve the relevancy of the selected item rules and to adapt hybrid recommendation process.

Towards this direction, a list of rule items of conceptually identical tags is identified, using the SWRL language [18], which is an inference engine of the determination of compatible rules. Simply run, the SWRL predicates compares the features service and the features device by comparing the feature value. If they are similar and its values, we inject rule item in user's agenda (see Algorithm 4). Specifically, we assume two available mobile devices of the current user domain were examined: one device having CPU speed equal to high (device 1) and another equal to low (device 2), resulting in 2.1 MHz and 0.5 MHz, respectively. A network of device 1 is {WiFi–3G}, while a device 1 supports: {WiFi–3G–5G}. To automatically check the available devices for running rule services correctly on the user domain, the SWRL rule (Algorithm 4) was employed. The devices requirements which have CPU speed, RAM size and network were checked with rule services features and then inject into the user's agenda when both service and rule items features are semantically matched.

---

**Algorithm 4: Check Device's Requirements with Situation Rule.**

---

User_Profile (?u) ∧ User_Domain (?u, ?d) ∧ hasCPUReq_device (?d, ?CPUSpeed_device) ∧
hasRAMReq_device (?d, ?RAMSize_device) ∧ hasNetworkReq_device (?d, ?network_device) ∧
Situation_Rule (?r, "Home_Intrusion_Alarm") ∧ hasCPUReq_rule (?rule, ?CPUSpeed_rule)
∧hasRAMReq_rule (?d, ?RAMSize_rule) ∧ hasNetworkReq_rule (?d, ?network_rule) ∧
greaterThan(?CPUSpeed_device, ?CPUSpeed_rule) ∧
greaterThan (?RAMSize_device, ?RAMSize_rule) ∧
greaterThan (?network_device, ?network_rule)
→ agenda(?u, ?r)

---

## 5. The Prototype

### 5.1. Prototype Implementation

We implemented a desktop prototype on NetBeans IDE and integrated it on Kali-Smart platform. The prototype may manage several customizable options according to user preferences. The ontology-based context-aware situation probabilistic recommendation process is the main module of our prototype, which is in charge of filtering and recommending suitable rules from the rule repository and then adapting these rules to available devices. The prototype enables a semantic matching between the current user context and the recommended situation rules based on the similarity function (Equation (6)), considering three-context similarities by using implicit and explicit preferences and Bayesian probability. Figure 5 shows the screenshots of main GUI.

The Java-based prototype can:

1. Create and save a novel profile with its preferences in the profile configuration file.
2. Manage any user's preferences, which has several customizable options according to user preferences. For example, the user can select between locations, roles to either worker or others. The weight for the recommendation processes can be also customized by the user, selecting a value among [0, 1] in the GUI.
3. Recommend relevant rules according to profile configuration.
4. Adapt actions services based on the device's characteristics information.

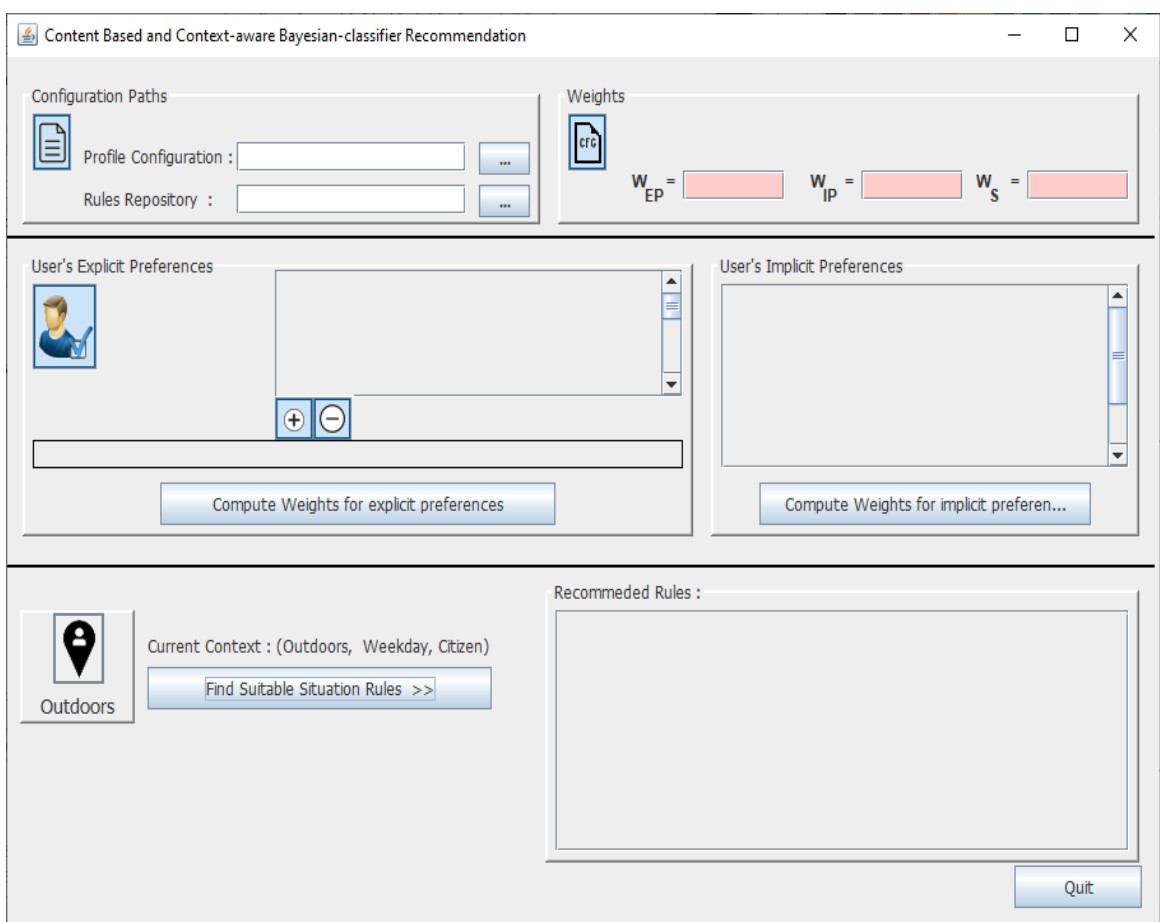

**Figure 5.** Main GUI of our recommendation and adaptation approach.

Figure 6 shows how the GUI specifies the preferences of a user and selects the requested values of his preferences. Indeed, the system calculates the weights of explicit attributes according to Equation (3) and selects the relevant situation rules with the best score among available situations rules. The score for each rule item is calculated based on Equation (6). Therefore, if the user changes its preferences, the system selects the new rules directly from the current context configuration set without performing phase 1.

The prototype is developed for situations enrichment and management by end-users and the administrator in smart environments. It allows an administrator to manage the rules repository by domain (*security*, *demotic*, *health*, *study*, *work*, *tourism* or *shopping*) and control the recommendation parameters according to user situation. It also helps end-users to profit from rich and dynamic user experiences.

Initially, when launching the prototype, the user should provide his social accounts (Facebook or Google Calendar) to enable the recommender to extract his/her interests and propose new situations rules. Since the user needs to exploit different available services in his smart domain, he has to define his preferences. After defining his preferences and daily situations, the recommender starts to suggest the user's relevant rules and appropriate services. To define the situations easily, we propose a mobile application GUI enabling to define daily situations and/or to subscribe to external sources for a richer and more dynamic application. The user can select a preferred device to run recommended services. To respond to user situations, the prototype needs to be aware of his context at all times and continues suggesting new situations rules according to his current location, preferences and needs.

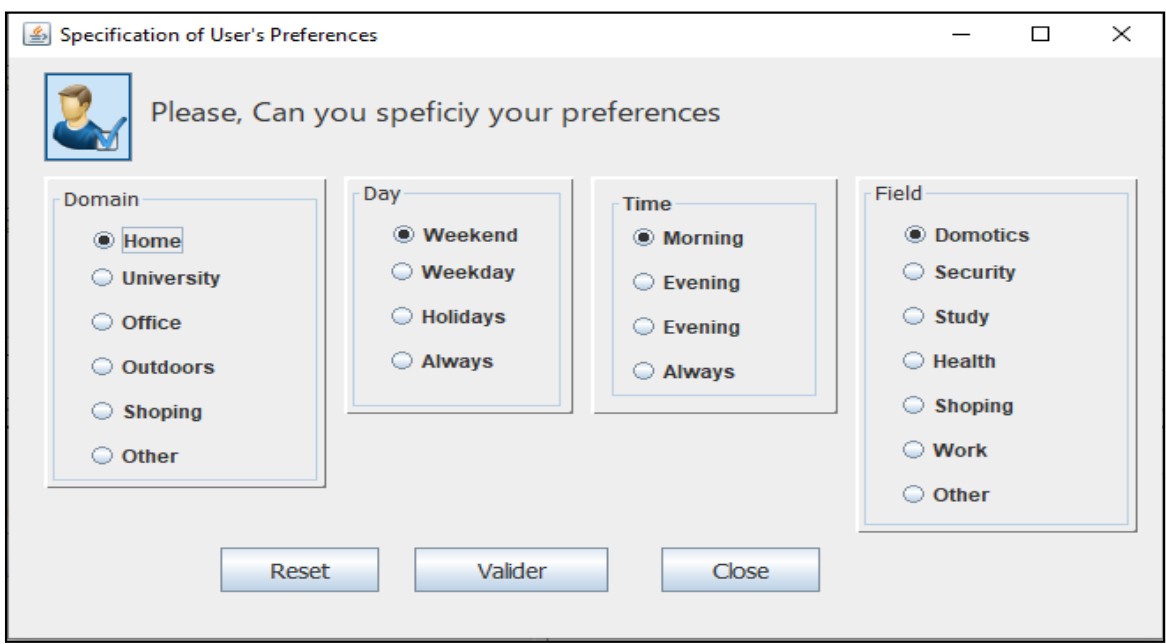

**Figure 6.** GUI for specifying the values of preferences.

*5.2. Possible Scenarios*

We used our prototype to assist the user in his daily life situations. We want to meet the user's needs in different places using the rule recommendation and service adaptation process based on the device's context information. We assume that the prototype has a registry of all his situation rules. As shown in Figure 7, the user has three domains (i.e., home, car and office domains).

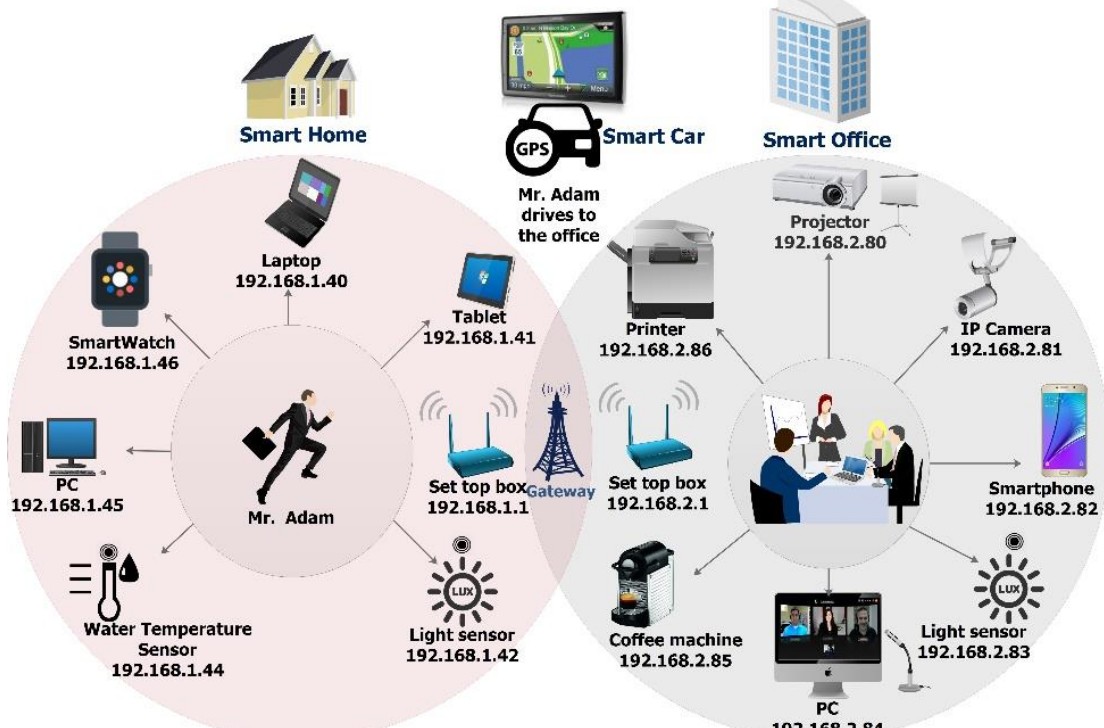

**Figure 7.** Possible scenarios.

If we assume that user's preferences are more important than situation, thus, preference is expressed with high priority and assigned to value ∈ [0.7, 1] situation is expressed with low priority and assigned to value ∈ [0, 0.3]). For example, $W_P = 0.7$ and $W_S = 0.3$. Since we have two types of preferences, so $W_{EP} = 0.35$ and $W_{IP} = 0.35$ as shown in Figure 8. These weights cannot be directly specified by the user, but we need an expert to prioritize such criteria. For the threshold between 0.6 and 1, we apply a threshold of 0.6. It permits to recommend wide range of situations rules to achieve high number of situation rules correctly recommended compared to other thresholds based on several test results.

**First Scenario.** The first scenario consists of responding to the user, Adam, who is at home. The system recommends suitable rule items according to the home's location, explicit and implicit preferences. The user sets his explicit preferences as shown in Figure 9. The PP vector (Preference Pair) of information presented in Section 4 are calculated, based on the provided profile configuration.

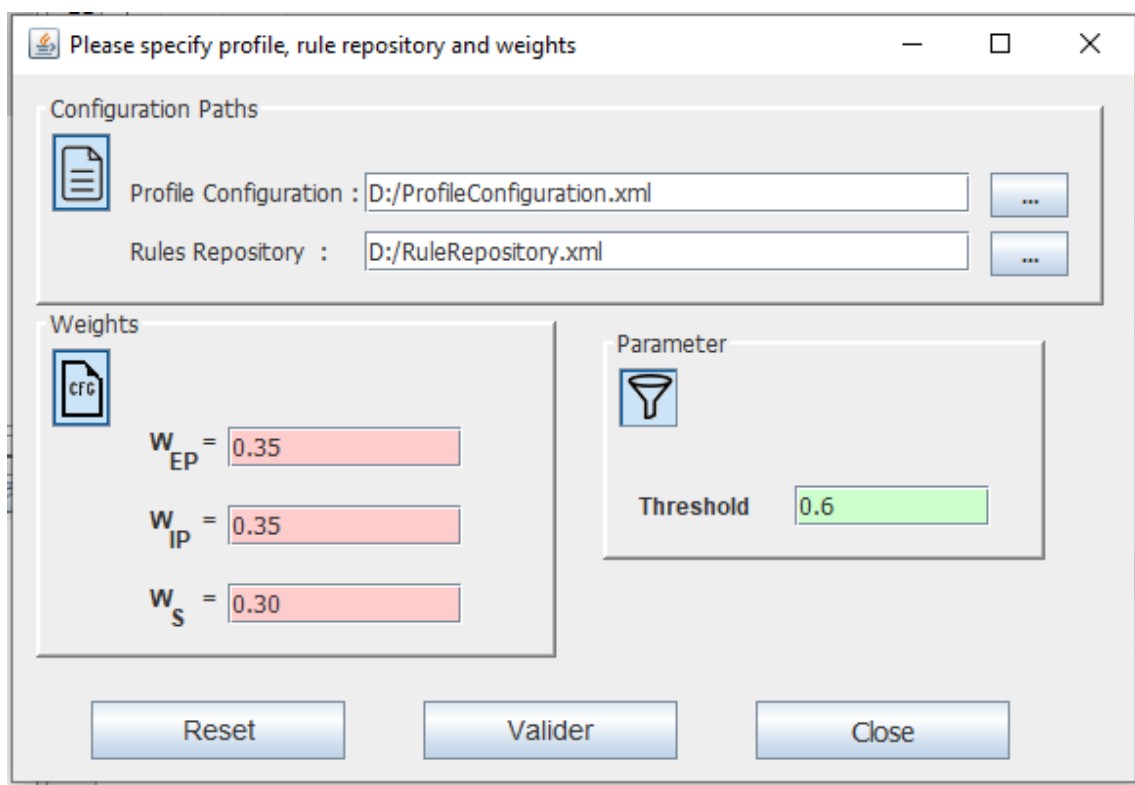

**Figure 8.** Similarity weights for all scenarios.

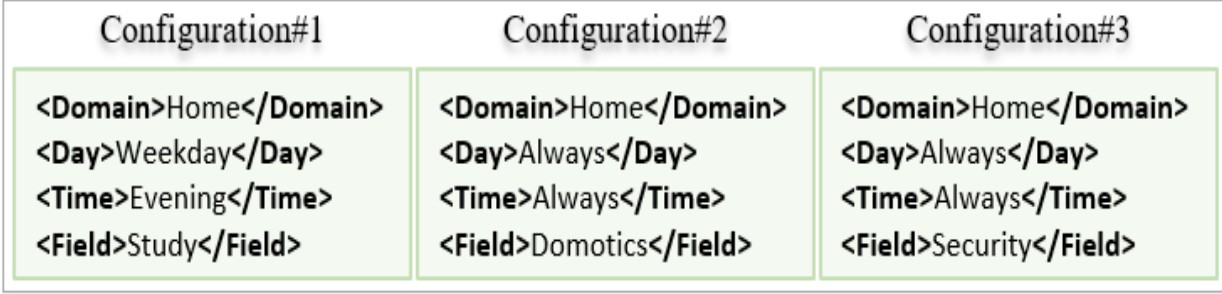

**Figure 9.** Explicit preferences for the first scenario.

Table 2 shows a list of relevant home situation rules (e.g., house_ intrusion_ alarm, house_ fire_ alarm, etc.) for every field (security, domotics and study), where each rule item $r_i$ is a recommended rule where the final score $Score_{r_i} \geq Thershold$. We pertain to

sorting the suitable rules by field according to the score reached through the ontology-based context-aware recommendation process. However, selecting the top-10 rules is sufficient. Moreover, due to the constrained resources of mobile devices, compatible rule items were automatically defined using SWRL language.

**Second Scenario.** A second scenario is when it is time to go to work; the system loaded the user's outdoors preferences to recommend new rule items and ensure the dynamic context changes (location is outside home and the available device is smartphone and car's navigator). The new recommended rules items are automatically updated and adapted to new current context. Table 3 shows a list of relevant home situation rules (e.g., *al_otahim_Market*, *car_map_weekday*, etc.) for every field (shopping and driving). Figure 10 illustrates the user explicit preferences for the second scenario.

**Table 2.** List of recommended rules with first preferences configuration and context (home).

| Field | Rule Item | $SIM_{EP}$ | $SIM_{IP}$ | $SIM_s$ | Score |
|---|---|---|---|---|---|
| Study | Study_Home_Evening | 0.8165 | 0.8135 | 0.7291 | 0.7892 |
| | Study_Home_Weekend | 0.8165 | 0.8135 | 0.6188 | 0.7561 |
| Domotics | Adjuste_Water_Temperature | 0.8165 | 0.8135 | 0.8184 | 0.8160 |
| | Morning_Preparation | 0.8165 | 0.8135 | 0.8184 | 0.8160 |
| | Morining_Turn_Light_Kitchen | 0.8165 | 0.8135 | 0.8184 | 0.8160 |
| | Enter_Bathroom_Light_On | 0.8165 | 0.8135 | 0.8184 | 0.8160 |
| | Wake_Up_All_Wakes_Up | 0.8165 | 0.8135 | 0.8184 | 0.8160 |
| | Leave_Bathroom_Light_Off | 0.8165 | 0.8135 | 0.8465 | 0.8244 |
| | Showing_Light_Color_Blue | 0.8165 | 0.8135 | 0.8465 | 0.8244 |
| Security | House_Fire_Alarm | 0.8165 | 0.8135 | 0.7326 | 0.7902 |
| | Home_Intrusion_Alarm | 0.8165 | 0.8135 | 0.6548 | 0.7669 |
| | Home_Room_Unlocking | 0.8165 | 0.8135 | 0.6231 | 0.7574 |
| | Security_Control_Door | 0.8165 | 0.8135 | 0.7326 | 0.7902 |
| | Night_Garage_Close | 0.8165 | 0.8135 | 0.5551 | 0.7370 |

**Table 3.** List of recommended rules with second preferences configuration and context (outdoors).

| Field | Rule Item | $SIM_{EP}$ | $SIM_{IP}$ | $SIM_s$ | Score |
|---|---|---|---|---|---|
| Driving | Path_Route | 0.4165 | 0.2235 | 0.5996 | 0.4038 |
| | Car_Map_Weekday | 0.4165 | 0.2235 | 0.3528 | 0.3298 |
| Shopping | Shop_Al_Othaim_Market | 0.4165 | 0.2235 | 0.4898 | 0.3709 |
| | Shop_Al_Basem_Shop | 0.4165 | 0.2235 | 0.4831 | 0.3689 |
| | Shoping_Order_Google_Calendar | 0.4165 | 0.2235 | 0.4048 | 0.3454 |
| | Shop_Open_Send_Email | 0.4165 | 0.2235 | 0.3072 | 0.3161 |

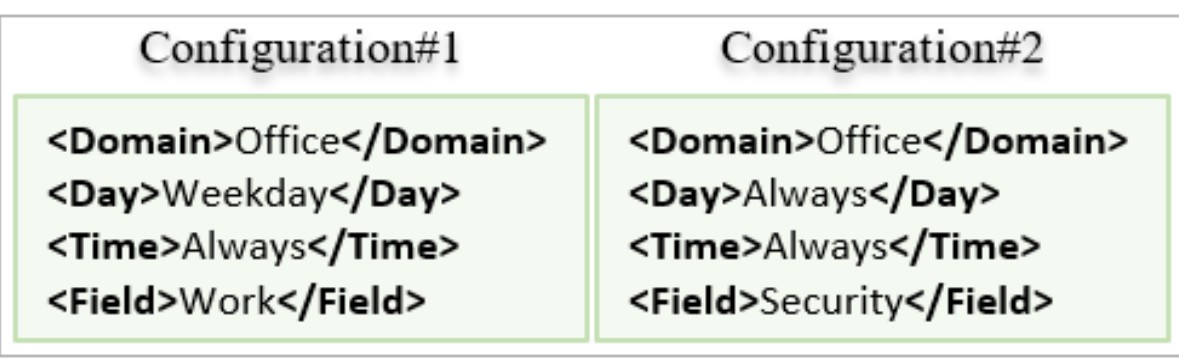

**Figure 10.** Explicit preferences for the second scenario.

**Third Scenario.** When a user enters his office, of course, his preference configurations will be loaded (see Figure 11). The system runs the ontology-based context-aware probabilistic recommendation process and the list of recommended rules is shown in Table 4. When meeting time comes, the service deployment module orchestrates the best way to adapt the interactive services after finding available devices. The system removes pen-click services on his desktop PC.

**Figure 11.** Explicit preferences for the third scenario.

**Table 4.** List of recommended rules with first preference configuration and context (home).

| Field | Rule Item | $SIM_{EP}$ | $SIM_{IP}$ | $SIM_s$ | Score |
|---|---|---|---|---|---|
| **Work** | Plugged_Device_Battery_Low | 0.8660 | 0.4780 | 0.2955 | 0.5590 |
| | Meeting_Office | 0.8660 | 0.4780 | 0.2899 | 0.5573 |
| | Office_High_Glucose_Inject_Insulin | 0.8660 | 0.4780 | 0.2610 | 0.5487 |
| | Work_Leave_Light_Off | 0.8660 | 0.4780 | 0.2610 | 0.5487 |
| | Save_Documents_After_Working_Time | 0.8660 | 0.4780 | 0.2351 | 0.5409 |
| | Mute_Phone_Enter_Work | 0.8660 | 0.4780 | 0.1117 | 0.5039 |
| **Security** | Turn_Camera_On_Leave_Office | 0.8660 | 0.4780 | 0.1619 | 0.5189 |
| | Office_Surveillance | 0.8660 | 0.4780 | 0.1241 | 0.5076 |

*5.3. Evaluation and Comparison*

5.3.1. Dataset

A data set of 100 rules for daily living of different fields (security, domotics, health, study and work) that help to evaluate the performance of semantic-based context-aware probabilistic recommendation approach. The specified values of user's preferences are randomly selected to automatically build various execution tests. We tested a series of

recommended rules items on the three preferences reconfiguration (home, outdoors and office). Figure 12 displays some rules items used in experimentation. For evaluation process, we include two strategies for feeding the dataset. The first one is the dataset generator module that creates a specified number of situations rules according to user abilities and needs of user in order to simulate the recommendation process of situations. The second one is the dataset injector module, which is responsible for enriching dataset with various data (location, time, date and tasks) of everyday situations from external sources using Netflix, Facebook, Twitter and Google Calendar. This data is extracted as text transformed in XML and injected into the user's application (rules repository).

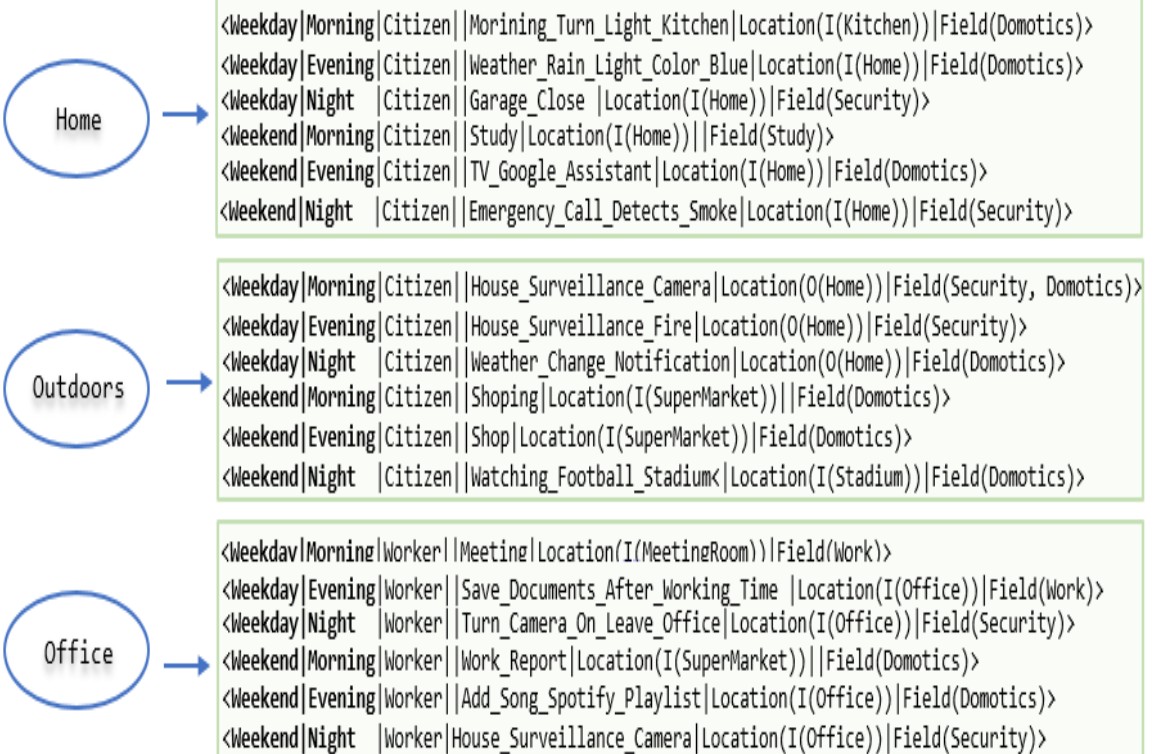

**Figure 12.** An example of a user rules repository in three user's domains.

5.3.2. Performance Measures

This section demonstrates how the proposed approach can achieve the recommendation accuracy based on user's context, his preferences and his available devices. The multi-domain rule repository (i.e., *history of users' rules*) of this experiment is used. We evaluate the performance in automatically selecting rules items in terms of:

- Precision (*P*) is the ratio of relevant recommended rule items to the user and the total among the recommended ones.
- Recall (*R*) is the ratio of relevant recommended items among the total number of all items.
- F1-score (*F1*) is used to evaluate a weighted average of precision and recall.

These metrics are calculated as follows:

$$P = \frac{TP}{TP + FP} \tag{7}$$

$$R = \frac{TP}{TP + FN} \tag{8}$$

$$F_1 = \frac{TP + TN}{TP + TN + FP + FN} \tag{9}$$

where:

- *TP* is the total number of relevant rules items (True Positive).
- *FP* is the total number of non-relevant rules items (False Positive).
- *TN* is the total number of relevant rules items that are not selected by the proposed approach (True Negative).
- *FN* is the total number of non-relevant rules items that are not selected by the proposed approach (False Negative).

### 5.3.3. Tests and Configuration Setup

We performed the following five tests:

- **Test # 1: Rules items recommendation based on explicit preference context**

We have applied the explicit preference context approach in order to extract the EP vector according to set of fields: application's fields (*"security"*, *"domotics"*, *"health"*, *"study"*, *"work"*, *"driving"* and *"shopping"*), user's domain ("home", "office", "university" and "outdoors"), day ("weekday" and "weekend") and time ("morning", "evening" and "night"). First, the system computes the weights for explicit preferences of a user according to Equation (1) for a profile configuration. Table 5 illustrates explicit weights for three profiles configurations.

**Table 5.** Explicit weights for three profiles configurations.

| Configurations | Smart Domain | Fields | Day | Time |
|---|---|---|---|---|
| Configuration#1 | Home | Study: 0.33 Domotics: 0.33 Security: 0.33 | Weekday: 0.33 Weekend: 0.67 | Weekday: 0.33 Weekend: 0.67 |
| Configuration#2 | Outdoors | Driving: 0.5 Shoping: 0.5 | Weekday: 1 | Weekday: 1 |
| Configuration#3 | Office | Work: 0.5 Security: 0.5 | Weekday: 1 | Weekday: 1 |

- **Test # 2: Rules items recommendation based on hybrid preference-based approach**

We applied both implicit and explicit preference approach to evaluate rule item according to current context of user. Hybrid preference-based recommendation compares location, day and time in a situation rule item with terms that define user preference to determine whether user likes that rule item.

- **Test # 3: Rules items recommendation based on implicit preference context**

We apply the content-based approach to extract the implicit preference vector according to the user's agenda history. This experiment allows us to compute the weights for implicit preferences of user according to Algorithm 2, and recommends relevant situation rules against the current user's context based on the rule item terms weights and rules items history (see Figure 13).

- **Test # 4: Rules items recommendation based on situation context Bayesian-based approach**

Based on the Bayesian approach, we generate $6 \times 4 \times 2 \times 3 = 120$ classes, and each class is represented by a set of weight vectors. Then, we use the situation context Bayesian-based approach to evaluate the probability of the rule item belonging to the situation context.

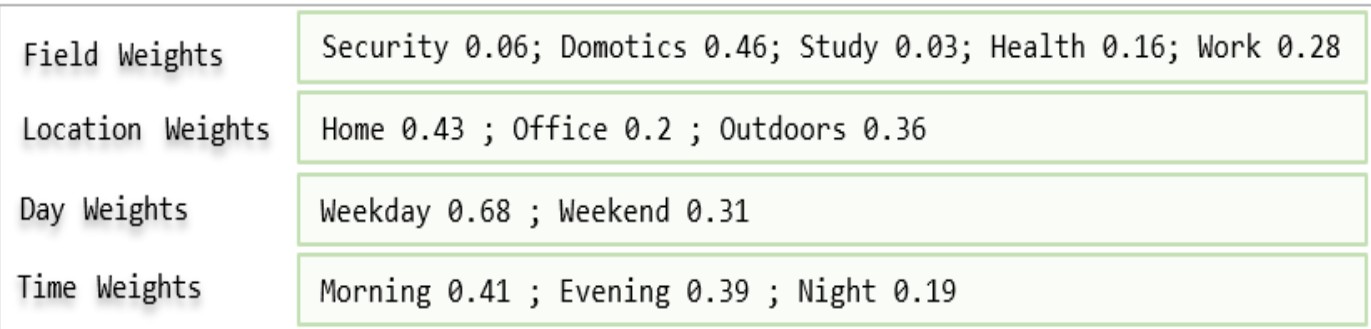

**Figure 13.** Implicit weights for rules history.

- **Test # 5: Rules items recommendation based on the proposed approach**

  In this test, we apply the context-aware semantic probabilistic approach to evaluate rule items according to users' preferences, user's situations and device capability.

  The performance measures of the proposed approach are evaluated and compared using three criteria: user's preferences, rule item content and context situation. The effectiveness of the semantic-based probabilistic context-aware approach was compared with some well-known classifiers (Decision Tree (DT), Support Vector Machine (SVM) and Random Forest (RF)). Note that all experiments have been conducted on Java Eclipse on an Intel processor core i5-2430 2.4 GHz and 8 GB RAM. The experiment was conducted on multi-smart domains dataset. This dataset consists of 100 rules items, 76 belong to daily activities and 24 belong to health and security situations rules. We divide dataset into two parts: training and testing. The first one uses the training data set with a size of 80% and the second has a size of 20%. The detail of configurations and situations rules items sets are presented in Table 6.

**Table 6.** Configuration setup.

| Parameter | Size |
|---|---|
| Number of situations rules items | 100 |
| Number of daily activities | 80 |
| Number of health and security rules | 20 |
| Number of profiles configurations | 3 |
| Number of available devices | 10–1000 |
| Weights Explicit Preference ($W_{EP}$) | 0.35 |
| Weights Implicit Preference ($W_{IP}$) | 0.35 |
| Weights Naive Bayes ($W_S$) | 0.30 |

*5.4. Results and Discussion*

5.4.1. Evaluation and Comparison Regarding Accuracy

The goal of this experiment is to analyze which recommendation criteria make the system more accurate when we rely on preference, rule items content or both. Thus, we have carried out several performance experiments using the proposed approach by observing five criteria:

1. Explicit Preferences (EP).
2. Implicit Preferences (IP).
3. Both Explicit and Implicit preferences (EIP).
4. Contextual-based Bayesian (Cx-Bayesian).
5. All criteria are considered (EIP and Cx-Bayesian).

Comparisons of accuracy across criteria (explicit preference, implicit preference, rule item content and situation context) are reported in Figure 14. The proposed approach reaches a promising precision based on all criteria as compared with other possible combinations of criteria of the proposed approach with 95.23% on rules repository. Besides,

the proposed approach achieves 96.42% recall and demonstrates a higher F1 score rate of 95.82%. In addition, even if implicit context content is extracted from the smart-domain dataset, the proposed approach performs better than other models.

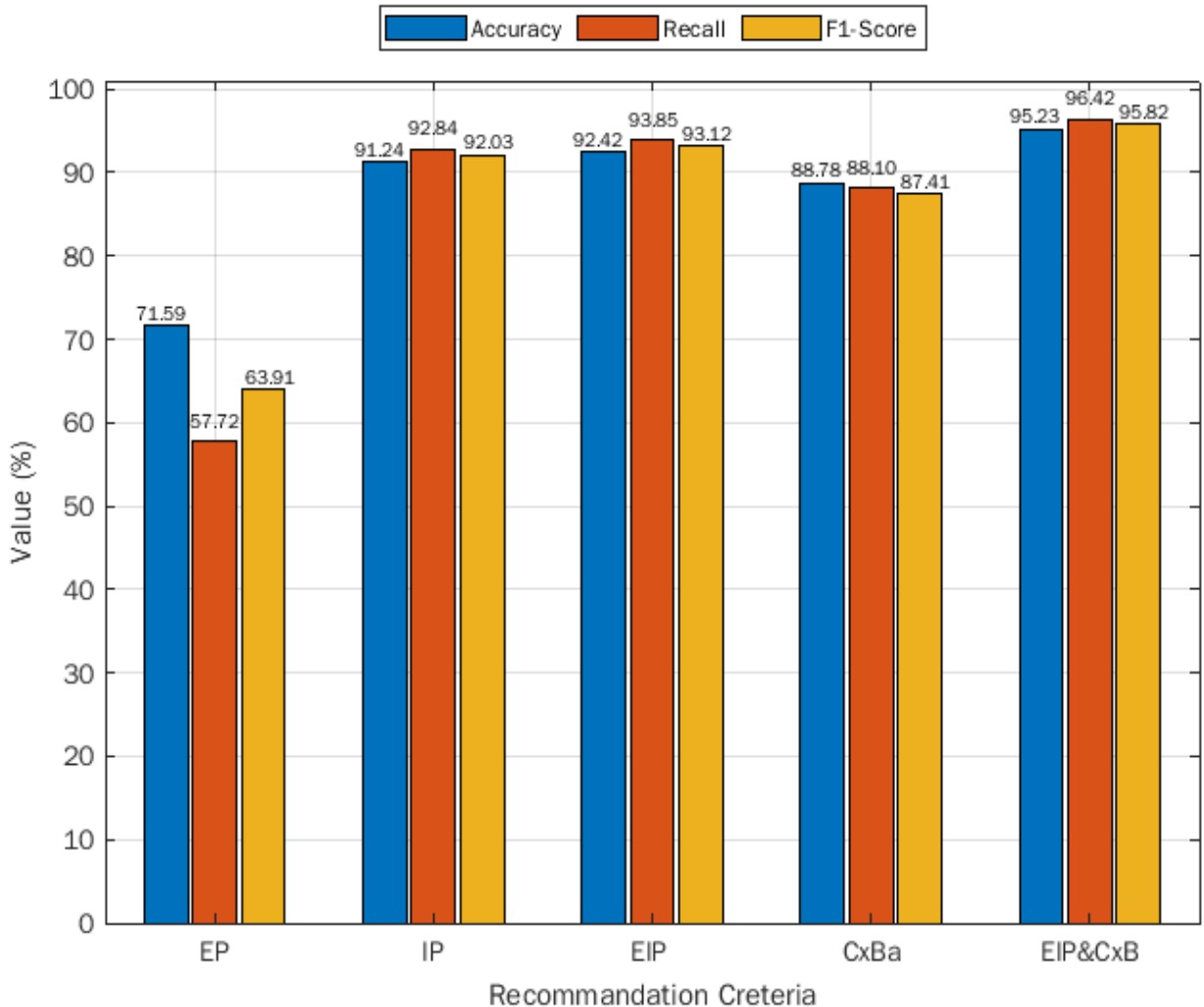

**Figure 14.** Comparison of accuracy for different criteria by the proposed approach.

### 5.4.2. Compared Classifiers Regarding Accuracy

Table 7 shows the comparison of the precision and recall of the proposed approach using non-contextual classifiers (DT, KNN and SVM). The proposed contextual Bayesian classifier based on Multi-OSCM ontology model attains a promising precision as compared to other non-contextual classifiers with 95.23% on multi-smart domain rules repository. Besides, the proposed approach demonstrates a higher recall of 95.82%. From these results, we can conclude that the proposed method is more effective than traditional classifiers in which ontology-based contextual Bayesian technique is applied.

**Table 7.** Comparison of the proposed approach with other classifiers.

| Classifier | Accuracy | Recall |
|---|---|---|
| KNN | 92.06% | 93.25% |
| DT | 80.95% | 86.39% |
| SVM | 87.30% | 86.39% |
| Proposed approach | 95.23% | 95.82% |

### 5.4.3. Evaluation of Response Time

In smart domains (home, city, health, tourism, etc.), the services recommendation and adaptation processes involve analyzing trade-offs between response time and number of mobile devices. Moreover, once a user's situation has been recommended it will be important to determine its applicability across the available devices. Thus, we evaluated the response time (e.g., total of recommendation and adaptation time) of the proposed approach by varying the numbers of mobile devices such that the number of situation rules is fixed to 80. As shown in Table 8, we observe that the increased number of mobile devices incurs service adaptation delay.

**Table 8.** Evaluation of response time (ms) of proposed approach using different criteria (number of situation rules is fixed to 80).

| Criteria | Number of Mobile Devices | | |
|:---:|:---:|:---:|:---:|
| | 10 | 50 | 100 |
| Explicit Preferences | 29.40 | 33.02 | 46.05 |
| Implicit Preferences (item content) | 25.20 | 36.74 | 61.18 |
| Explicit and Implicit Preferences | 42.32 | 51.37 | 84.19 |
| Contextual Bayesian (context situation) | 148.56 | 153.08 | 193.27 |
| Preferences and Contextual Bayesian | 177.04 | 182.24 | 245.14 |

### 5.4.4. Discussion

For achieving a good recommendation system, we include implicit context preference based on Algorithm 2 in the learning and recommendation process, since when rule items repository increases, so does accuracy. That algorithm was proposed with the hypothesis that the users cover all situation rules, and then enriches his agenda with new relevant situation rule for daily living not already doing it. With that hypothesis, our system gives mainly relevant with close matched context ignoring other possible contextual configurations.

The comparative analysis of recommendation approaches used for situation rules recommendation reveals the following important points:

- Among the four recommendation criteria, the proposed approach-using user's implicit and explicit preferences, user's history, user's current situation gives highest accuracy for rule item recommendation. However, explicit preference-based approach gives lowest accuracy.
- In addition, the proposed approach shows highest level of performance with 100% smart domains covered under the different user's domains (security, demotic, health, study, work, tourism and shopping). The explicit preference-based approach shows lowest area covered under poor user-specified preferences.
- Among the four approaches, explicit preference-based approach presents the fastest algorithm for rule item recommendation with the highest prediction speed and lowest training time. However, the proposed approach presents best recommendation approach with the highest accuracy and recall rates and acceptable execution time.

The proposed approach on context-aware preferences and rules items contents attains promising accuracy results as compared to traditional machine learning approaches as showed in Figure 14 and Table 7. However, the semantic-based context-aware probabilistic approach is much better than traditional machine learning approaches in managing and recommending relevant rules in terms of accuracy, recall and F1-score. The response time of our approach on daily life rules dataset is demonstrated in Table 8 towards the recommendation of relevant rules from three user's locations (home, office and outdoors). It is semantically flexible in situation enrichment and adaptation model for context-aware pervasive applications. However, the rules recommendation time must be improved in large datasets.

In our future work, will consider the optimum execution time of the large number of user situations datasets from the real-world daily life situations whiles considering the collaborative mechanisms and high-performance machine, which makes the results more practical.

## 6. Conclusions

In this paper, we proposed and evaluated a dynamic and modular methodology for classification and recommendations of user's situation rules for adaptable context-aware mobile applications. As presented, the proposal mainly consists of two processes, the learning process and the recommendation process. The former can be summarized in the semantic classification of user's situations rules. While the latter is based on the dynamic injection of situations rules at run-time through the hybridization of the content-based approach, the Bayesian classifier approach and the ontology-based SWRL rules. The recommendation process can be applied to any smart domain and use explicit and implicit user preferences rules and Bayesian classifier to increase performances for relevant situation rules recommendation. Besides, the use of the ontology model on learning and recommendation processes facilitates the filtering and customization of situation rules. The approach has been demonstrated through a simple case study to simulate its overall execution flow. The experimental evaluation of accuracy and performance through a large dataset reflects the effectiveness of our approach in comparison with existing works. Future works directions include improvements into similarity and inference in matching tools and similarity measures with different types of multimedia properties.

**Author Contributions:** Conceptualization, A.L., P.R. and A.A.; methodology, A.L., P.R. and A.A.; software, A.L. and A.A.; validation, A.L., P.R. and A.A.; formal analysis, A.L. and A.A.; investigation, A.L., P.R. and A.A.; resources, A.L., P.R. and A.A.; data curation, A.L., P.R. and A.A.; writing—original draft preparation, A.L.; writing—review and editing, P.R. and A.A.; visualization, A.L. and A.A.; supervision, P.R. and A.A.; project administration, A.A. All authors have read and agreed to the published version of the manuscript.

**Funding:** Not applicable.

**Institutional Review Board Statement:** Not applicable.

**Informed Consent Statement:** Not applicable.

**Data Availability Statement:** Data available on request due to restrictions.

**Conflicts of Interest:** The authors declare no conflict of interest.

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
