# Peer review of "Novel Semantic-Based Probabilistic Context Aware Approach for Situations Enrichment and Adaptation"

_applsci, doi:10.3390/app12020732_

Round 1
Reviewer 1 Report
The authors have fixed and responded all issues I have raised in the previous review. Congratulations! I recommend to accept this manuscript.
Author Response
Responses to reviewers comments :
(Our responses are highlighted in red and also in the main manuscript)
First of all, I would like to thank the reviewers for accepting reviewing our paper and also for their valuable comments. Please find below our responses to all the comments point by point:
Reviewer 1 Comments:
The authors have fixed and responded all issues I have raised in the previous review.
Congratulations! I recommend to accept this manuscript.
I would like to express my respect and my gratitude for accepting reviewing our paper.

Reviewer 2 Report
In the paper present a new semanti-based probabilistic systems that enriches and adapts dynamically user situations rules in vast domains. The rules have to fit eith users preferences, user current situations, and devices capabilities.
The proposed system has two main process the learning process and recomendation process.
Authors present develop a prototype with this model and evaluate it.
For developers of context-aware systems, what they propose in this article is interesting. However, the way it is presented is a bit confusing or poorly structured.
How is the prototype developed in JAVA used?
Who would use this prototype for, isn't it clear?
In section 5.3.1. How was the dataset generated, is it your own or was it generated from others?
explain in detail.
I would recommend adding a discussion section as the paper has several contributions listed and the authors would be expected to conclude on these. As well as mention the limitations of your proposed methodology, if any.
there are typo errors, for instance, page 29, line 862.
It is recommended to review the format of the magazine, since several texts are in different formats and apparently font size. For example, equations, table titles, rule codes, etc.
I Identify several repeating references in the article, for instance reference 17, 18 and 29 is the same. Also references 27 and 28 or 31 y 33. Therefore, authors are recommended to review all their references in the text. This in order to verify that there are no missing or excess references in the document. For instance, reference 34 does not appear listed.
Figures are mislabeled. As I read the document, I identified two figures number 9 within the document. View page 23 line 716 and page 22 line 698. The same situations happens with figure 11, see pages 25 and 26.
You have to define the acronyms used in the article, for example, SWRL (Semantic Web Rule Language).
Author Response
Responses to reviewers comments :
(Our responses are highlighted in red and also in the main manuscript)
First of all, I would like to thank the reviewers for accepting reviewing our paper and also for their valuable comments. Please find bellow our responses to all the comments point by point:
Reviewer 2 Comments:
- How is the prototype developed in JAVA used?
The java prototype is developed for situations enrichment and management by end-users and the administrator in smart environments. It allows an administrator to manage the rules repository by domain (security, demotic, health, study, work, tourism, shopping) and control the recommendation parameters according to user situation. It also helps end-users to profit from rich and dynamic user experiences.
- Who would use this prototype for, isn't it clear?
Initially, when launching the prototype, the user should provide his social accounts (Facebook or Google Calendar) to enable the recommender to extract his interests and propose new situations rules. Since the user needs to exploit different available services in his smart domain, he has to define his preferences. After defining his preferences and daily situations, the recommender starts to suggest the user’s relevant rules and appropriate services. To define the situations easily, we propose a mobile application GUI enabling to define daily situations and/or to subscribe to external sources for a richer and more dynamic application. The user can select a preferred device to run recommended services. To respond to user situations, the prototype needs to be aware of his context at all times and continues suggesting new situations rules according to his current location, preferences and needs.
- In section 5.3.1. How was the dataset generated, is it your own or was it generated from others? explain in detail.
For evaluation process, we include two strategies for feeding the dataset. The first one is the dataset generator module that creates a specified number of situations rules according to user abilities and needs of user in order to simulate the recommendation process of situations. The second one is the dataset injector module, which is responsible for enriching dataset with various data (location, time, date and tasks) of everyday situations from external sources using Netflix, Facebook, Twitter and Google Calendar. This data is extracted as text transformed in XML and injected into the user’s application (e.g.rules repository). We have carefully revised and enhanced the section Section 5.3.1.
- I would recommend adding a discussion section as the paper has several contributions listed and the authors would be expected to conclude on these. As well as mention the limitations of your proposed methodology, if any.
Yes, we agree.
Discussions and limitations of the proposed work are added and detailed in Section 5.4.4.
- There are typo errors, for instance, page 29, line 862.
Fixed
- It is recommended to review the format of the magazine, since several texts are in different formats and apparently font size. For example, equations, table titles, rule codes, etc.
The format of the paper, tables and figures are revised according to the format of the magazine.
- I Identify several repeating references in the article, for instance reference 17, 18 and 29 is the same. Also references 27 and 28 or 31 y 33. Therefore, authors are recommended to review all their references in the text. This in order to verify that there are no missing or excess references in the document.
Corrected
- For instance, reference 34 does not appear listed.
It was a mistake from our part; corrected.
- Figures are mislabeled. As I read the document, I identified two figures number 9 within the document. View page 23 line 716 and page 22 line 698. The same situations happens with figure 11, see pages 25 and 26.
Fixed.
- You have to define the acronyms used in the article, for example, SWRL (Semantic Web Rule Language).
All acronyms used in the article are defined.

Round 2
Reviewer 2 Report
Thanks for listening to my suggestions.
This manuscript is a resubmission of an earlier submission. The following is a list of the peer review reports and author responses from that submission.
Round 1
Reviewer 1 Report
- The authors should explain more what is Multi-OCSM and how it has been used in your work.
- In line 424-425, what is the list of semantic context data relates to user's role, user's location extracted from WordNet. If possible please give the clear example of this classification.
- How do the authors define the suitable hyperparameters: Wep, Wip and Ws in equation (6) in line 583. As seen in Figure 8. the authors used Wep=0.35, Wip=0.35, and Ws=0.3. How to pick these hyperparameters.
- Again, Thershold in line 666, the authors used 0.6. How does the authors define this value? Does it has any scientific soundness about this value?
- Evaluation and comparison regarding accuracy section (line762-769), what algorithm did the authors use in each approach? for example, contented-based, there are many works in this approach. Have the authors run these approaches with the same conditions e.g. dataset(n=100)? Please explain more about these experiments. How do you split train and test dataset?
- I recommend the bar graph in Figure12 should put the value text on the top of each bar because content-based, hybrid-based and proposed approach gives very close values.
- what is SWRL rules? please cite the referenced paper if possible (first see in line 809)
- Since, there are many supervised machine learning approaches, why the authors use the Bayesian Classifier? have you considered alternative classifiers?
- Actually, there are also several similarity measures, why the author choose the cosine similarity?
Reviewer 2 Report
In this paper, the authors present an approach to improve recommendation systems using context awareness.
Although the subject seems relevant, the quality of writing is very poor, especially grammar, which makes the paper incomprehensible. For instance, we can note sentences such as
- line 30, "The recommendation is arguably one of the most important issues in such applications due to possible serve users in daily life"
- line 50, "However, these techniques are not efficient in practice while extracting a large amount of metadata from crowdsourcing, a variety of,..., users' needs"
- Line 217, "It recommends places where users have not visited before"
- Line 585, "0<Wep<0", and similar definition at lines 586 and 587
- Line 757. The sentence is even not finished.
In addition, the problem the authors are addressing is not very clear. It seems, they take an example of an intelligent agenda to assist people in planning their tasks, but I have to admit I was not able to clearly what the authors are demonstrating. They introduce a lot of terms such as "situation", "rule item", "situation recommendation, "situation rule" which are only defined in a formal way in Section 3 and 4, but do not have a concrete meaning. All the terms that are heavily used throughout the paper should be clearly explained since the introduction as well as the use case which is being developed. Indeed, the agenda application does not seem very convincing to implement such complex system. So, the authors either better explain motivations behind this use case or explain another application with important challenges that could be solved by their solution.
Finally, evaluation methodology is not clear. How were the authors able to evaluate the accuracy? Did they perform tests with some of their colleagues?
In conclusion, I recommend to the authors to deeply review their article in order to
- explain the motivations behind their works and state the application they are addressing
- make the paper accessible to readers that may not have all the background on the fields of the authors